# Adult Pancreatoblastoma: Clinical Insights and Outcomes Compared to Pancreatic Ductal Adenocarcinoma (PDAC)

Han Yin [1,†], Fernanda Romero-Hernandez [2,†], Amir Ashraf Ganjouei [2], Jaeyun Jane Wang [2], Audrey Brown [2], Kenzo Hirose [3], Ajay V. Maker [3], Eric Nakakura [3], Carlos Corvera [3], Kimberly S. Kirkwood [3], Alexander Wilhelm [4], June S. Peng [3], Adnan Alseidi [3] and Mohamed A. Adam [3,*]

[1] School of Medicine, University of California, San Francisco, CA 94143, USA; han.yin@ucsf.edu
[2] Department of Surgery, University of California, San Francisco, CA 94143, USA; maria.romerohernandez@ucsf.edu (F.R.-H.)
[3] Department of Surgery, Division of Surgical Oncology, University of California, San Francisco, CA 94143, USA; carlos.corvera@ucsf.edu (C.C.)
[4] Department of Visceral Surgery, Clarunis—University Center for Gastrointestinal and Liver Diseases, St. Clara Hospital and University Hospital Basel, 4058 Basel, Switzerland
[*] Correspondence: mohamed.adam@ucsf.edu
[†] These authors contributed equally to this work.

**Abstract:** Pancreatoblastoma is perceived to be aggressive in adults; however, data are limited due to the rarity of the disease. We benchmarked clinico-pathologic characteristics, outcomes, and survival of adult patients with pancreatoblastoma to a comparable PDAC cohort using the National Cancer Database (NCDB). This study included 301,204 patients: 35 with pancreatoblastoma and 301,169 PDAC patients. Pancreatoblastoma patients were younger than PDAC patients (56 vs. 69 years, $p < 0.001$). More pancreatoblastoma patients were managed at academic institutions (63.0% vs. 40.7%, $p = 0.047$). The most frequent primary site was the head and the neck of the pancreas. There were no differences in tumor size (4.2 cm vs. 3.7 cm, $p = 0.828$), lymph node positivity (14.3% vs. 26.4%, $p = 0.103$), or metastasis at time of diagnosis (31.4% vs. 46.1%, $p = 0.081$). The majority of pancreatoblastoma patients underwent resection compared to a minority of PDAC patients (69.7% vs. 15.5%, $p < 0.001$). Time from diagnosis to surgery was longer for pancreatoblastoma patients (33 vs. 14 days, $p = 0.030$). Pancreaticoduodenectomy was the most common type of resection in the pancreatoblastoma and PDAC groups (47.8% vs. 67.7%, $p = 0.124$). Among resected patients, pancreatoblastoma patients were less likely to receive radiation (4.8% vs. 37.0%, $p = 0.002$), but the use of chemotherapy was similar to PDAC patients (60.9% vs. 70.7%). After matching, median overall survival was longer for pancreatoblastoma than PDAC (59.8 months vs. 15.2 months, $p = 0.014$).

**Keywords:** pancreatoblastoma; PDAC; outcomes

## 1. Introduction

Pancreatoblastoma is a rare malignant neoplasm originating from the pancreatic epithelial exocrine cells. It is most frequently seen in infants and children [1–3], with an annual incidence of 0.004 per 100,000 population of all ages [4]. Adult pancreatoblastoma is exceptionally rare and typically presents with non-specific symptoms related to tumor mass effect, such as abdominal pain, weight loss, and jaundice [4]. Radiographic findings of pancreatoblastoma are also non-specific and have been described in the literature as heterogeneous, contrast-enhancing lesions with well-defined margins [5]. Ultimately, definitive diagnosis depends primarily on pathologic findings through the presence of characteristic squamoid corpuscles [6,7]. The mainstay of treatment is radical surgical resection [8]. However, in general, pancreatoblastoma in adults is thought to be associated with more aggressive tumor behavior and shorter survival compared to pediatric patients [4]. Most

cases appear to be sporadic, although there is emerging evidence that pancreatoblastoma may be associated with Familial Adenomatous Polyposis (FAP) in adults [9].

The literature on pancreatoblastoma is sparse. To date, there have been limited case series published specifically on adult pancreatoblastoma. The published reports are limited by the use of older data dating back to the 1980s or a lack of important clinico-pathologic information such as tumor size [6,10]. Importantly, while pancreatoblastoma is perceived to be more aggressive than PDAC in adults [11–16]. No studies have directly compared patient characteristics and outcomes between the two conditions. Thus, the aim of our study was to describe demographic, clinico-pathologic, and treatment characteristics and survival of adult patients with pancreatoblastoma using a large database and to compare it to those diagnosed with PDAC.

## 2. Materials and Methods

### 2.1. Cohort Selection and Data Source

This was a retrospective study using the National Cancer Database (NCDB) from 2004 to 2019. The NCDB is a nationwide database sponsored by the American Cancer Society and the American College of Surgeons that contains clinical oncology data sourced from 1500 Commission on Cancer–accredited cancer programs in the United States [9]. Data were coded according to the Commission on Cancer Registry Operations and Data Standards Manual, the American Joint Committee for Cancer (AJCC) Manual for Staging of Cancer, and the International Classification of Disease for Oncology (ICD-O-3).

The NCDB pancreas-Participant User Files was queried for all adult patients (age ≥18 years) with the histologic diagnosis of pancreatoblastoma (ICD-O-3 code: 8971) and pancreatic adenocarcinoma (ICD-O-3 code: 8140). The following demographic variables were extracted: patient age, sex, race, ethnicity, year of diagnosis, insurance status, facility type, and facility location. The following clinical variables were extracted: Charlson–Deyo Comorbidity Index, primary tumor site, tumor size (based on pathology if available, otherwise determined radiographically), tumor extension, lymphovascular invasion (based on pathology), clinical TNM staging, diagnostic procedure, treatment received (radiotherapy, chemotherapy, and surgical resection), and time to surgical resection. The following outcomes were also extracted: surgical margins, lymph-node involvement, 90-day mortality, 30-day readmission, and hospital length of stay. Due to the de-identified nature of the dataset, this study was granted exemption status by our Institutional Review Board.

### 2.2. Statistical Analysis

Descriptive statistics of patient demographic and clinical characteristics were performed. After stratifying by histology type, a propensity match was performed using a nearest-neighbor algorithm to create a 1:3 matched cohort (pancreatoblastoma to PDAC). The cohort was matched using the following variables: patient age, race, sex, and clinical TNM stage. The Wilcoxon rank-sum test for continuous variables and the Pearson $\chi^2$ test for categorical variables were used to compare baseline characteristics and outcomes between the two groups. A Kaplan–Meier survival analysis was performed for all matched patients as well as matched patients who underwent surgery. A *p*-value of 0.05 was considered statistically significant. Statistical analyses were performed using R v4.1.3 (R Foundation for Statistical Computing, Vienna, Austria).

## 3. Results

### 3.1. Adult Patients with Pancreatoblastoma and PDAC

A total of 301,204 patients were identified, of whom 35 were pancreatoblastoma patients, and 301,169 were PDAC patients (Table 1). Pancreatoblastoma patients were younger than PDAC patients (median 56 vs. 69 years, *p* < 0.001). There were no significant differences in sex, race, or Hispanic ethnicity between the two groups. A larger proportion of pancreatoblastoma patients were treated at academic centers compared to PDAC patients (63.0% vs. 40.7%, *p* = 0.047). The most frequent primary site was the head and the neck

of the pancreas for both the pancreatoblastoma and PDAC groups (42.9% vs. 53.2%, $p = 0.287$). The median tumor size was similar between the pancreatoblastoma and PDAC groups (4.2 cm vs. 3.5 cm, $p = 0.828$). Lymphovascular invasion was more often present in pancreatoblastoma (22.9% vs. 4.7%, $p < 0.001$). No pancreatoblastoma patients presented with clinical T4 disease, compared to 21.2% of PDAC patients that presented with clinical T4 disease ($p < 0.001$). Pancreatoblastoma patients more often underwent surgical resection compared to PDAC patients (69.7% vs. 15.5%, $p < 0.001$).

**Table 1.** Demographic and clinical characteristics of adult patients with pancreatoblastoma and pancreatic ductal adenocarcinoma.

| | Pancreatoblastoma (*n* = 35 [1]) | PDAC (*n* = 301,169 [1]) | Pre-Match *p*-Value [2] | Post-Match *p*-Value [2] |
|---|---|---|---|---|
| Age (Years) | 56 (42-61) | 69 (61–77) | <0.001 | 0.895 * |
| Sex | | | 0.483 | 0.824 * |
|     Male | 20 (57.1) | 154,234 (51.2) | | |
|     Female | 15 (42.9) | 146,935 (48.8) | | |
| Race | | | 0.414 | 0.565 * |
|     White | 27 (77.1) | 251,891 (83.6) | | |
|     Black | 7 (20.0) | 37,994 (12.6) | | |
|     Other | 1 (2.9) | 11,284 (3.7) | | |
| Hispanic Ethnicity | | | 0.854 | 0.777 |
|     Non-Hispanic | 31 (93.9) | 273,093 (94.7) | | |
|     Hispanic | 2 (6.1) | 15,406 (5.3) | | |
| Charlson-Deyo Score | | | 0.044 | 0.187 |
|     0–1 | 35 (100.0) | 270,012 (89.7) | | |
|     ≥2 | 0 (0.0) | 31,157 (10.3) | | |
| Year of Diagnosis | | | 0.915 | 0.974 |
|     2004–2008 | 9 (25.7) | 76,303 (25.3) | | |
|     2009–2014 | 13 (37.1) | 121,669 (40.4) | | |
|     2015–2019 | 13 (37.1) | 103,197 (34.3) | | |
| Insurance Type | | | 0.005 | 0.583 |
|     Private | 20 (58.8) | 91,670 (31.1) | | |
|     Government | 14 (41.2) | 191,219 (64.8) | | |
|     Other | 0 (0.0) | 12,263 (4.2) | | |
| Facility Type | | | 0.047 | 0.121 |
|     Academic | 17 (63.0) | 121,786 (40.7) | | |
|     Community | 0 (0.0) | 15,388 (5.1) | | |
|     Comprehensive/Integrated | 10 (37.0) | 161,888 (54.1) | | |
| Facility Location | | | 0.644 | 0.544 |
|     Northeast | 8 (29.6) | 64,683 (21.6) | | |
|     South | 8 (29.6) | 109,446 (36.6) | | |
|     Midwest | 8 (29.6) | 77,705 (26.0) | | |
|     West | 3 (11.1) | 47,228 (15.8) | | |
| Primary Site | | | 0.287 | 0.792 |
|     Head and Neck | 15 (42.9) | 160,075 (53.2) | | |
|     Body and Tail | 9 (25.7) | 77,776 (25.8) | | |
|     Other | 11 (31.4) | 63,318 (21.0) | | |

**Table 1.** *Cont.*

| | Pancreatoblastoma (*n* = 35 [1]) | PDAC (*n* = 301,169 [1]) | Pre-Match *p*-Value [2] | Post-Match *p*-Value [2] |
|---|---|---|---|---|
| Tumor Size (cm) | 4.2 (3.5,6.1) | 3.7 (2.8, 4.8) | 0.828 | 0.849 |
| Tumor Extension | | | 0.357 | 0.357 |
| Intrapancreatic | 9 (47.4) | 60,145 (32.4) | | |
| Extrapancreatic | 6 (31.6) | 66,996 (36.1) | | |
| Vascular | 4 (21.1) | 58,214 (31.4) | | |
| Lymphovascular Invasion | | | <0.001 | 0.908 |
| Present | 8 (22.9) | 14,092 (4.7) | | |
| None | 7 (20.0) | 27,323 (9.1) | | |
| Unknown | 20 (57.1) | 259,692 (86.2) | | |
| Clinical T Stage | | | <0.001 | 0.514 * |
| cT0 | 1 (2.9) | 1270 (0.4) | | |
| cT1 | 1 (2.9) | 12,694 (4.2) | | |
| cT2 | 7 (20.0) | 68,735 (22.8) | | |
| cT3 | 6 (17.1) | 75,668 (25.1) | | |
| cT4 | 0 (0.0) | 63,785 (21.2) | | |
| Unknown | 20 (57.1) | 79,017 (26.2) | | |
| Clinical N Stage | | | 0.103 | 0.512 * |
| cN0 | 30 (85.7) | 221,529 (73.6) | | |
| cN1 | 5 (14.3) | 79,640 (26.4) | | |
| Clinical M Stage | | | 0.081 | 0.614 * |
| cM0 | 24 (68.6) | 162,280 (53.9) | | |
| cM1 | 11 (31.4) | 138,889 (46.1) | | |
| Surgery | | | <0.001 | <0.001 |
| No Surgery | 10 (30.3) | 248,403 (84.5) | | |
| Surgery | 23 (69.7) | 45,627 (15.5) | | |

[1] No. (%); median (IQR); percentages may not add up to 100% due to missing data. [2] Statistical analyses performed using Chi$^2$ test for categorical variables. * Denotes variables used for matching. PDAC, pancreatic ductal adenocarcinoma.

### 3.2. Patients Who Underwent Resection

We next performed a subgroup analysis of patients who underwent surgical resection. There was a total of 23 pancreatoblastoma patients and 45,627 PDAC patients who underwent surgical resection. Demographic and clinical characteristics of these resected pancreatoblastoma and PDAC patients are summarized in Appendix A. Among the resected patients, there were more pancreatoblastoma patients who had distant metastatic disease compared to PDAC patients (26.1% vs. 3.6%, *p* < 0.001). We examined factors related to receipt of surgical and non-surgical treatments (Table 2). Wait time between diagnosis and resection was significantly longer for pancreatoblastoma patients vs. PDAC patients (39 days vs. 22 days, *p* = 0.002). Pancreaticoduodenectomy was the most commonly performed procedure for both groups (47.8% vs. 67.7%, *p* = 0.124). Receipt of chemotherapy was similar between the two groups (60.9% vs. 70.7%, *p* = 0.300). Radiation treatment was used significantly less to treat pancreatoblastoma patients (4.8% vs. 37.0%, *p* = 0.002).

**Table 2.** Summary of treatments for adult patients with pancreatoblastoma and pancreatic ductal adenocarcinoma who underwent definitive surgical resection.

| | Pancreatoblastoma (n = 23 [1]) | PDAC (n = 45,627 [1]) | *p*-Value [2] |
|---|---|---|---|
| Clinical M Stage | | | <0.001 |
| cM0 | 17 (73.9) | 43,974 (96.4) | |
| cM1 | 6 (26.1) | 1653 (3.6) | |
| Chemotherapy | | | 0.300 |
| Yes | 14 (60.9) | 32,261 (70.7) | |
| No | 9 (39.1) | 13,366 (29.3) | |
| Radiation | | | 0.002 |
| Yes | 1 (4.8) | 15,726 (37.0) | |
| No | 20 (95.2) | 26,772 (63.0) | |
| Time to Surgery (Days) | 39 (0,175) | 22 (4,58) | 0.002 |
| Extent of Surgery | | | 0.124 |
| Whipple | 11 (47.8) | 30,906 (67.7) | |
| Partial pancreatomy | 6 (26.1) | 5990 (13.1) | |
| Total pancreatectomy [2] | 5 (21.7) | 5929 (13.0) | |
| Other | 1 (4.3) | 2802 (6.1) | |

[1] No. (%); median (IQR); percentages may not add up to 100% due to missing data. [2] Statistical analyses performed using Chi$^2$ test for categorical variables. PDAC, pancreatic ductal adenocarcinoma.

### 3.3. Matched Patients Who Underwent Resection

Finally, we performed a subgroup analysis of patients who underwent surgery from the propensity matched cohort. From our initial matched cohort of 128 patients, there were 22 pancreatoblastoma patients and 40 PDAC patients who underwent surgical resection (Table 3). Regarding postoperative outcomes, there were no significant differences in median length of stay (7 days vs. 9 days, *p* = 0.765), 30-day readmission (13.6% vs. 15.0%, *p* = 0.552), or 90-day mortality (9.5% vs. 3.0%, *p* = 0.310). Oncologically, there were no significant differences in positive surgical margins (10.5% vs. 21.1%, *p* = 0.325) or lymph node positivity (36.8% vs. 62.9%, *p* = 0.067).

Resected pancreatoblastoma patients trended toward a longer median survival than resected PDAC patients (71.7 months vs. 16.5 months, *p* = 0.150) (Figure 1). We then examined our entire matched cohort and saw that overall pancreatoblastoma patients had a significantly longer survival than PDAC patients (59.8 months vs. 15.2 months, *p* = 0.014, Figure 1). Relative survival of pancreatoblastoma patients by resection and presence of metastatic disease is presented in Appendix B.

**Table 3.** Summary of outcomes for adult patients with pancreatoblastoma and pancreatic ductal adenocarcinoma who underwent definitive surgical resection.

| | Pancreatoblastoma (n = 22 [1]) | PDAC (n = 40 [1]) | *p*-Value [2] |
|---|---|---|---|
| Lymph Nodes | | | 0.067 |
| Negative | 12 (63.2) | 13 (37.1) | |
| Positive | 7 (36.8) | 22 (62.9) | |
| Surgical Margins | | | 0.325 |
| Positive | 2 (10.5) | 8 (21.1) | |
| Negative | 17 (89.5) | 30 (78.9) | |

**Table 3.** *Cont.*

|  | Pancreatoblastoma (n = 22 [1]) | PDAC (n = 40 [1]) | *p*-Value [2] |
|---|---|---|---|
| 90-Day Mortality |  |  | 0.310 |
|     Alive | 19 (90.5) | 32 (97.0) |  |
|     Dead | 2 (9.5) | 1 (3.0) |  |
| 30-Day Readmission |  |  | 0.552 |
|     Yes | 3 (13.6) | 6 (15.0) |  |
|     No | 19 (86.4) | 34 (85.0) |  |
| Length of Stay (Days) | 7 (6,11) | 9 (6,14) | 0.765 |

[1] No. (%); median (Q1,Q3); percentages may not add up to 100% due to missing data. [2] Statistical analyses performed using Chi$^2$ test for categorical variables. PDAC, pancreatic ductal adenocarcinoma.

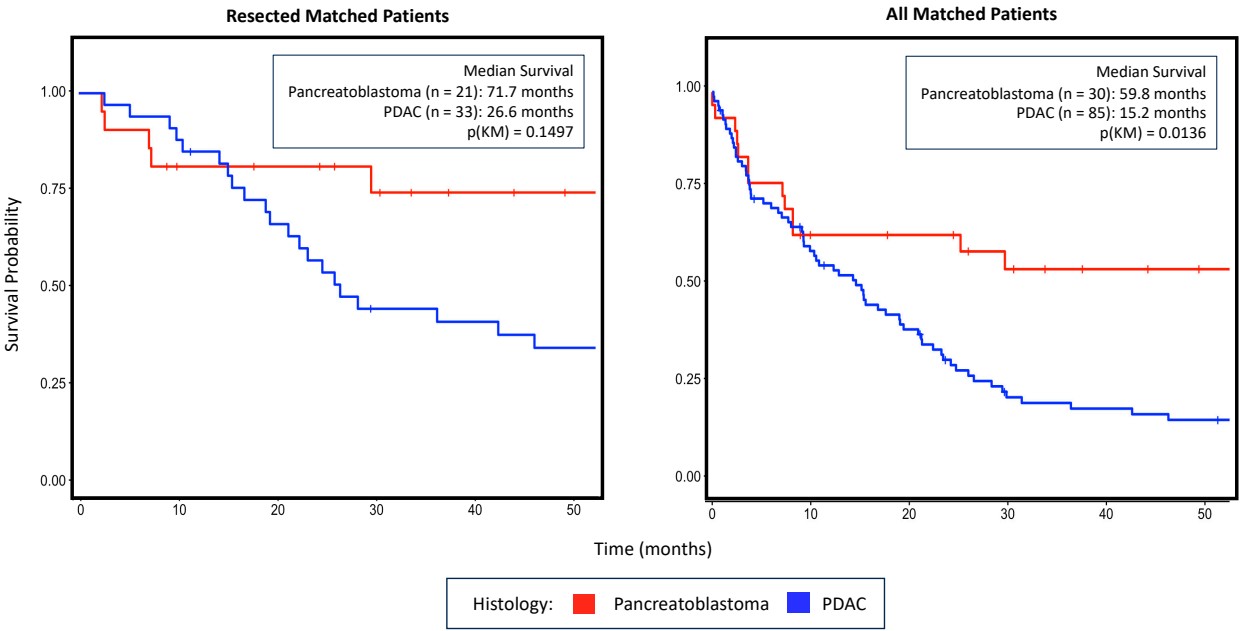

**Figure 1.** (**Left**) Kaplan–Meier survival curves for the same matched cohort subset on only patients that underwent surgery. (**Right**) Kaplan–Meier survival curves of patients from a matched a cohort of pancreatoblastoma and PDAC patients that had survival data available. PDAC, pancreatic ductal adenocarcinoma.

### 3.4. Treatment Trends

Given the lack of consensus in the literature on the optimal treatments for pancreatoblastoma, we generated an overview of the treatments received by the 35 pancreatoblastoma patients in the NCBD stratified by resection status. In order to benchmark the treatment trends we observed in our cohort to previously published cases, we generated a comparison of the demographics, clinical characteristics, and treatment of the pancreatoblastoma patients in our cohort and the 40 patients reported in an adult pancreatoblastoma literature review published by Zouros et al. [15] in 2015. Results are shown in Appendix C.

### 4. Discussion

This study is the largest contemporary series to characterize clinico-pathologic characteristics, treatment, and outcomes of adult pancreatoblastoma patients, with a relevant comparison to adult PDAC patients. Pancreatoblastoma patients were younger than PDAC patients. However, there were no differences between pancreatoblastoma and PDAC in tumor size, lymph node positivity, or presence of metastasis at the time of diagnosis. The most frequent primary site is the head and the neck of the pancreas and the majority of

pancreatoblastoma patients underwent pancreaticoduodenectomy. The majority received chemotherapy at rates comparable to PDAC patients; however, after matching, overall survival was significantly longer for pancreatoblastoma patients compared to PDAC patients.

An unexpected clinical finding was the similar tumor size (median 4.2 cm) in the pancreatoblastoma and the PDAC cohorts. This observation contradicts prior studies that reported pancreatoblastoma to be larger on average [6,17]. Zouros et al. [15] examined 40 cases from 1986 to 2015 and reported a median tumor size of 7.3 cm. The difference in sizes between our study and Zouros et al.'s report could be reflective of improvements in diagnostic technology leading to earlier detection, as these previous reports include cases from as early as 1986, while the earliest cases in our study were from 2004.

Eleven (31.4%) pancreatoblastoma patients in our study had metastatic disease at diagnosis, which is fewer than what was reported in the prior literature (34–59%) [6,8]. However, previously published reports included both patients with metastatic disease at diagnosis and patients that developed metastatic disease at a later time, which could explain the difference in the frequency we observed. In our study, pancreatoblastoma patients had a lower rate of metastasis at diagnosis than PDAC patients (46.1%). This supports the notion that pancreatoblastoma may be less aggressive than PDAC.

In our study, the majority of pancreatoblastoma patients (62.0%) underwent pancreatic resection even in the presence of metastatic disease (26.1%). This contrasts with the 3% metastatic PDAC patients who underwent resection. One possible reason that pancreatoblastoma patients frequently undergo resection is because of lack of consensus on how to treat them. Furthermore, there are several case reports that describe long-term survival after resection for pancreatoblastoma patients with metastatic disease to the liver [13,17,18]. Notably, these reports do not offer any additional insight into why long-term survival is achievable for pancreatoblastoma patients—even in cases of metastatic disease. Conversely, the lower rate of resection in metastatic PDAC patients could be due to better established standards of care for metastatic PDAC patients and evidence to support alternative treatments such as radiation and chemotherapy.

The majority of patients in our cohort (62.9%) received chemotherapy, which is also comparable with what has been previously reported (Appendix B). When stratified by resection status, we see that 75.0% of unresected patients underwent chemotherapy and 60.9% of resected patients received chemotherapy. However, there was significant variation in timing of treatment for resected patients. Specifically, nine patients received neoadjuvant chemotherapy, four received adjuvant chemotherapy, and one received both (Figure 1). This is likely due to the lack of consensus regarding the optimal management of pancreatoblastoma; thus, usage tends to be highly institution-dependent [19]; however, there appears to be a definite role for chemotherapy in some capacity, especially in cases of metastatic pancreatoblastoma. For example, a 2020 case series of four patients in Germany with metastatic pancreatoblastoma reported adequate tumor control with Oxaplatin-containing treatment regimens [20]. This is consistent with pediatric pancreatoblastoma, where there is expert consensus on the beneficial role of chemotherapy [21].

In our matched cohort, we saw pancreatoblastoma patients had a significantly longer time between diagnosis and surgery. This increase is likely a manifestation of the rarity of the disease and limited knowledge and evidence in the management of adult pancreatoblastoma. Oncologic and postoperative outcomes of patients with pancreatoblastoma were overall similar to those with PDAC.

Earlier studies reported much shorter survival times for pancreatoblastoma. The first literature review of pancreatoblastoma in 1995 by Klimstra et al. [22], which included five cases from 1986 to 1995 reported a poor median survival of only 18 months for adult patients. The 2015 report by Zouros et al. included 40 patients from 1986 to 2016 and recorded date of death or last contact. Although Zouros et al. did not calculate the median survival, our own analysis of their data showed a median survival of 41 months. In contrast, our contemporary cohort of matched patients had a longer median survival of 71.7 months.

Taken together, this suggests that adult pancreatoblastoma is less aggressive than initial reports seem to indicate that survival has improved over time.

Survival is also significantly longer for pancreatoblastoma compared to PDAC, which only had a median survival of 10.6 months in our matched cohort, which also indicates that pancreatoblastoma is less aggressive than PDAC. This is not surprising when we consider the underlying genetic differences between the two malignancies. Key driver mutations found in PDAC such as KRAS, TP53, and CDKN2A[23] are absent from pancreatoblastoma. Unlike PDAC, pancreatoblastoma is associated with mutations in the Wnt/Beta-catenin signaling pathway [23]. Although the overall number of cases remains small, we argue that a trend that pancreatoblastoma is relatively less aggressive than PDAC has emerged. PDAC no longer appears to be a good benchmark for pancreatoblastoma, and there is a need for us to shift away from this conception.

Possible limitations of our study include the small sample size of the pancreatoblastoma cohort. Even with propensity matching, the comparison may still be affected by random sampling bias due to the degree of down sampling peformed in the PDAC group. The analysis is retrospective in nature, with the possibility of coding errors and selection bias. Additionally, the use of a large national database in which many different centers contributed cases over an extended period of time also leads to additional variability in how variables are defined and reported. Despite these limitations, given the rarity of pancreatoblastoma, we believe the NCBD is the appropriate source of data for this extremely rare disease with limited published data.

This relatively large and contemporary study enhances our understanding of demographic and clinico-pathologic characteristics and outcomes of adult patients with pancreatoblastoma. We showed that patients with pancreatoblastoma have similar clinical characteristics and postoperative outcomes compared to PDAC patients. The patterns of presentation and treatment for pancreatoblastoma have evolved with patients now presenting with smaller tumors and less metastatic disease at presentation. While adult pancreatoblastoma has been previously conceptualized as a uniformly aggressive malignancy, our findings show a good portion of adult pancreatoblastoma patients present with metastatic disease but may have better survival times compared the PDAC patients. This raises the question of whether pancreatoblastoma patients are being over treated for pancreatoblastoma and if we need to reconsider how we treat these patients. Although we are unable to draw strong conclusions due to the rarity of this disease and the limitations of our study design, we hope that benchmarking pancreatoblastoma against PDAC in this report will give providers a more a more nuanced view of the behavior of pancreatoblastoma and help inform decision making for their patients.

## 5. Conclusions

This study describes the largest and most contemporary series examining clinico-pathologic features and outcomes of adult patients with pancreatoblastoma using a large database. We also compared the outcomes of patients with pancreatoblastoma to those with PDAC using a propensity-matched algorithm. Systemic chemotherapy was used frequently, similar to its use in PDAC; however, overall survival was significantly longer in pancreatoblastoma compared to PDAC. Given the limitations in sample size, further research is warranted to better understand the disease course and optimal treatment of this rare malignancy.

**Author Contributions:** Conceptualization, H.Y., F.R.-H., J.J.W., A.A.G., A.B., J.S.P., A.W., K.H., E.N., C.C., K.S.K., A.V.M., A.A. and M.A.A.; methodology, H.Y., F.R.-H., J.J.W., A.A.G. and M.A.A.; software, A.A.G.; validation, H.Y. and F.R.-H.; formal analysis. H.Y., F.R.-H., J.J.W. and A.A.G.; investigation, H.Y., F.R.-H., J.J.W. and A.A.G.; resources, M.A.A.; data curation H.Y., F.R.-H., J.J.W. and A.A.G.; writing—original draft preparation, H.Y., F.R.-H., J.J.W., A.A.G. and A.B.; writing—review and editing, all authors; visualization, H.Y., F.R.-H., J.J.W., A.A.G. and A.B.; supervision, M.A.A.; project administration, H.Y., F.R.-H., J.J.W., A.A.G., A.B. and M.A.A. All authors have read and agreed to the published version of the manuscript.

**Funding:** J.J.W. was supported by the UCSF Noyce Initiative for Digital Transformation in Computational Biology and Health, Computational Innovator Postdoctoral Fellowship Award.

**Institutional Review Board Statement:** This study was granted exemption status by our Institutional Review Board based on the de-identified nature of the dataset (UCSF IRB approval number is 22-36919).

**Informed Consent Statement:** Patient consent was waived based on the de-identified nature of the dataset.

**Data Availability Statement:** The datasets generated during and/or analyzed during the current study are not publicly available but are available from the National Cancer Database on reasonable request.

**Conflicts of Interest:** The authors declare no conflicts of interest. The funders had no role in the design of this study; in the collection, analyses, or interpretation of data; in the writing of the manuscript; or in the decision to publish the results.

## Appendix A

**Table A1.** Demographic and clinical characteristics of resected adult patients with pancreatoblastoma and pancreatic ductal adenocarcinoma.

| | Pancreatoblastoma (n = 23 [1]) | PDAC (n = 45,627 [1]) | *p*-Value [2] |
|---|---|---|---|
| Age (Years) | 56 (45–61.5) | 67 (59–74) | <0.001 |
| Sex | | | 0.911 |
| Male | 12 (52.2) | 23,274 (51.0) | |
| Female | 11 (47.8) | 22,353 (49.0) | |
| Race | | | 0.034 |
| White | 16 (69.6) | 39,427 (86.4) | |
| Black | 6 (26.1) | 4561 (10.0) | |
| Other | 1 (4.3) | 1639 (3.6) | |
| Hispanic Ethnicity | | | 0.979 |
| Non-Hispanic | 20 (95.2) | 41,724 (95.1) | |
| Hispanic | 1 (4.8) | 2143 (4.9) | |
| Charlson-Deyo Score | | | 0.139 |
| 0–1 | 23 (100.0) | 41,661 (91.3) | |
| ≥2 | 0 (0.0) | 3966 (8.7) | |
| Year of Diagnosis | | | 0.465 |
| 2004–2008 | 6 (26.1) | 12,077 (26.5) | |
| 2009–2014 | 7 (30.4) | 18,800 (41.2) | |
| 2015–2019 | 10 (43.5) | 14,750 (32.3) | |
| Insurance Type | | | 0.738 |
| Private | 10 (45.5) | 16,776 (37.4) | |
| Government | 12 (54.5) | 26,377 (58.9) | |
| Other | 0 (0.0) | 585 (1.3) | |
| Facility Type | | | 0.021 |
| Academic | 15 (78.9) | 21,428 (47.4) | |
| Community | 0 (0.0) | 1533 (3.4) | |
| Comprehensive/Integrated | 4 (21.1) | 22,267 (49.2) | |

**Table A1.** *Cont.*

| | Pancreatoblastoma (n = 23 [1]) | PDAC (n = 45,627 [1]) | *p*-Value [2] |
|---|---|---|---|
| Facility Location | | | 0.436 |
| Northeast | 6 (31.6) | 9255 (20.5) | |
| South | 8 (42.1) | 16,900 (37.4) | |
| Midwest | 4 (21.1) | 12,283 (27.2) | |
| West | 1 (5.3) | 6790 (15.0) | |
| Primary Site | | | 0.116 |
| Head and Neck | 12 (52.2) | 32,610 (71.5) | |
| Body and Tail | 7 (30.4) | 7808 (17.1) | |
| Other | 4 (17.4) | 5209 (11.4) | |
| Tumor Size (cm) | 4.0 (3.4, 6.8) | 3.2 (2.5, 4.2) | 0.695 |
| Tumor Extension | | | 0.016 |
| Intrapancreatic | 7 (46.7) | 8534 (25.5) | |
| Extrapancreatic | 4 (26.7) | 20,906 (62.4) | |
| Vascular | 4 (26.7) | 4078 (12.2) | |
| Lymphovascular Invasion | | | 0.805 |
| Present | 8 (34.8) | 11,822 (25.9) | |
| None | 6 (26.1) | 12,584 (27.6) | |
| Unknown | 9 (39.1) | 21,220 (46.5) | |
| Clinical T Stage | | | <0.001 |
| cT0 | 1 (4.3) | 100 (0.2) | |
| cT1 | 1 (4.3) | 3959 (8.7) | |
| cT2 | 5 (21.7) | 11,340 (24.9) | |
| cT3 | 6 (26.1) | 12,707 (27.8) | |
| cT4 | 0 (0.0) | 2347 (5.1) | |
| Unknown | 10 (43.5) | 15,174 (33.3) | |
| Clinical N Stage | | | 0.828 |
| cN0 | 19 (82.6) | 36,878 (80.8) | |
| cN1 | 4 (17.4) | 8749 (19.2) | |
| Clinical M Stage | | | <0.001 |
| cM0 | 17 (73.9) | 43,974 (96.4) | |
| cM1 | 6 (26.1) | 1653 (3.6) | |

[1] No. (%); median (IQR); percentages may not add up to 100% due to missing data. [2] Statistical analyses performed using Chi$^2$ test for categorical variables. PDAC, pancreatic ductal adenocarcinoma.

## Appendix B

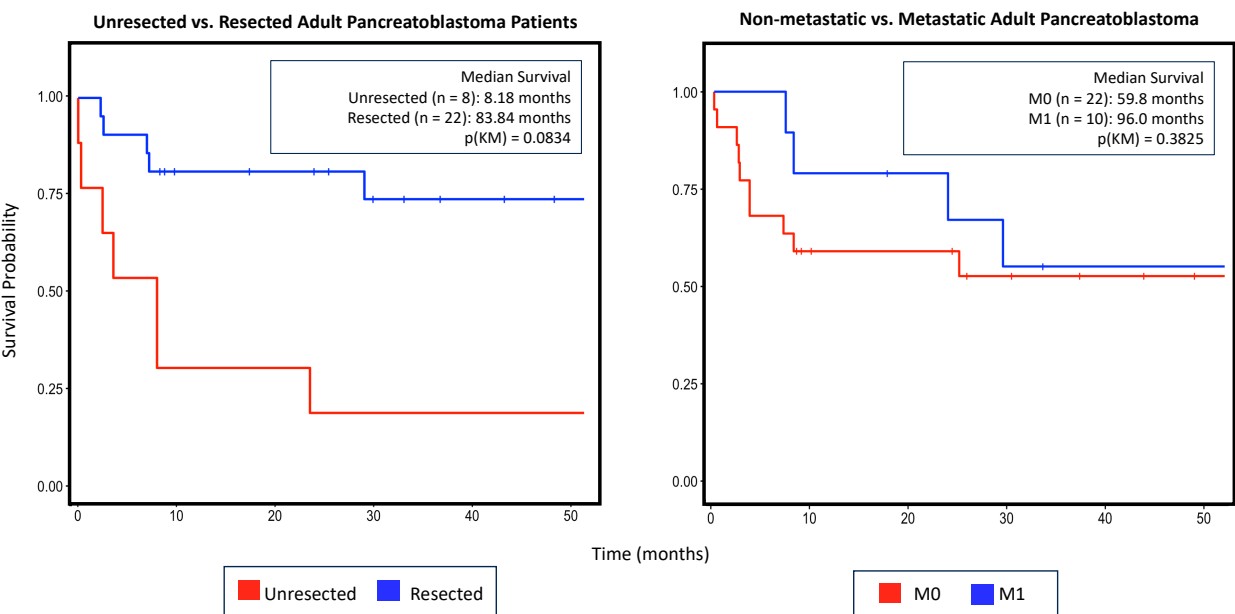

**Figure A1.** (**Left**) Kaplan–Meier survival curves of unresected and resected adult pancreatoblastoma patients that had survival data available. (**Right**) Kaplan–Meier survival curves of adult pancreatoblastoma patients with non-metastatic (M0) and metastatic disease (M1) based on clinical staging with available survival data.

## Appendix C

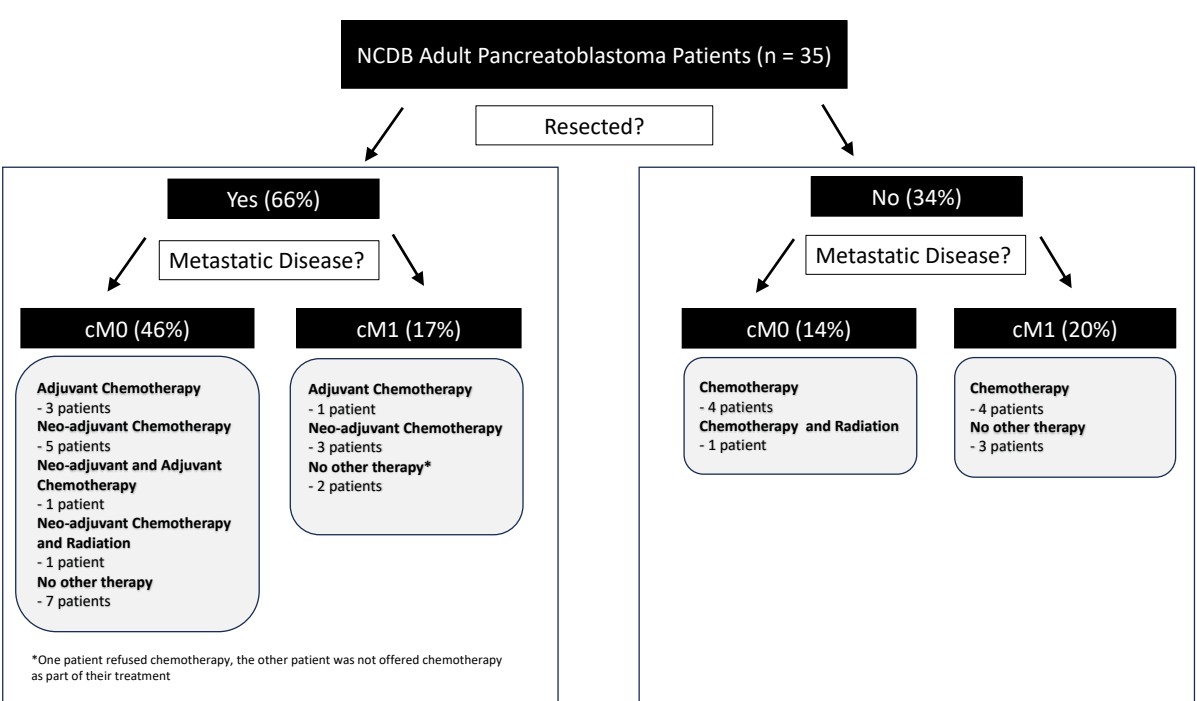

**Figure A2.** Summary of non-surgical treatments (chemotherapy and/or radiation) received by adult patients in the NCDB pancreatoblastoma cohort stratified by patients that underwent resection with or without metastatic disease at time of diagnosis. NCDB—National Cancer Database.

**Table A2.** Comparison of adult pancreatoblastoma patients in the NCDB1 from 2004 to 2019 and the Zouros et al. [14] review on adult pancreatoblastoma cases from 1986 to 2015.

| | NCDB (n = 23 [1]) | Zouros et al. (n = 38 [1,2]) | *p*-Value [3] |
|---|---|---|---|
| Age (Years) | 56 (45–61.5) | 67 (59–74) | <0.001 |
| Sex | | | 0.911 |
| Male | 20 (57.1) | 17 (44.7) | |
| Female | 15 (42.9) | 21 (55.3) | |
| Tumor Size (cm) | 4.2 (3.5, 6.1) | 7.3 (4.5, 9.0) | 0.034 |
| Metastatic Disease [4] | | 10 (26.3) | |
| Yes | 11 (31.4) | 28 (62.7) | |
| No | 24 (68.6) | | 0.979 |
| Chemotherapy | | 17 (44.7) | |
| Yes | 22 (62.9) | 21 (55.3) | |
| No | 13 (37.1) | | 0.139 |
| Radiation [5] | | 7 (18.4) | |
| Yes | 2 (6.2) | 31 (71.6) | |
| No | 30 (93.8) | | 0.465 |
| Resection | | 33 (86.8) | |
| Yes | 23 (69.7) | 5 (13.2) | |
| No | 10 (30.3) | 10 (26.3) | |
| Survival (Months) | 59.8 (8.81,) | 41 (30,) | |

[1] No. (%); median (Q1,Q3); (95% LCL—UCL). Percentages may not add up to 100% due to missing data. [2] Two cases were excluded from this review due to missing data. [3] Statistical analyses performed using $Chi^2$ test for categorical variables. [4] For the NCBD patients, metastatic disease is defined by clinical M staging, while in Zouros et al., patients were considered to have metastatic disease if presence of metastasis was described in their report. [5] Patients that underwent chemo-radiation in the Zouros et al. report we noted as having had both chemotherapy and radiation in this table. NCDB, National Cancer Database. PDAC, pancreatic ductal adenocarcinoma.

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
