# Peer review of "Adult Pancreatoblastoma: Clinical Insights and Outcomes Compared to Pancreatic Ductal Adenocarcinoma (PDAC)"

_curroncol, doi:10.3390/curroncol31090370_

Round 1
Reviewer 1 Report
Comments and Suggestions for Authors
Summary
In this retrospective propensity matched study, a large public cancer database was used to compare adult pancreatoblastoma patients with PDAC patients. The authors did not find any significant differences in the location, tumor size, lymph node status or metastasis status at the time of diagnosis. They however, found that the time to surgery was significantly longer for pancreatoblastoma patients compared to PDAC patients. They also found that the overall survival rate for resected patients was much longer in pancreatoblastoma patients compared to PDAC patients.
Strengths
Large sample size.
Comments
Introduction
1. Please mention the usual demographic profile of pancreatoblastoma and PDAC patients along with the available data about clinical features of the tumors, setting a stage to introduce your study.
Results
1. Can you add a supplementary table with information about each of the pancreatoblastoma patients included in the study?
2. Can you also evaluate using your data the difference in survival between resected and non-resected pancreatoblastoma patients; pancreatoblastoma with metastasis versus those without.
3. Do you have information about the specific chemotherapy used in pancreatoblastoma patients?
Reviewer 2 Report
Comments and Suggestions for Authors
The authors describe the outcomes of 35 patients diagnosed with pancreatoblastoma from the National Cancer Database comparing this with the outcomes for pancreatic ductal adenocarcinoma patients. Although NCDB is a useful took to study rare diseases however there is potential for significant bias in drawing any conclusion from a series of only 35 patients treated in many different institutions. The manuscript tries to draw some conclusions by comparing this group with PDAC patients however nothing really novel is presented here. Some comments to help the authors improve their manuscript:
1) Abstract. Need to specify what patients are we looking here....are these resected patients or data such as size are radiologic? also some numbers make no sense "the majority (69.7% vs 15,5%) underwent resection...what are these percentages?
2) the potential for bias is significant...even a few misclassified patients as pancreatoblastoma can divert to completely wrong conclusions in a cohort of only 23 patients who underwent resection
3) Materials and methods. needs better definitions for example is tumor size radiologic or pathologic? did the authors selected patients that had only a minimum follow up? did they exclude cases that were found on autopsy etc etc. I suggest they read NCDB papers that describe extensively their methods
4) statistical analysis. it doesnt make sense why the authors decided to match the patients for demographics and clinical stage and then they included a variable from pathology...lymphovascular invasion
5) the discussion is reasonable however all these studies suffer from the data not being granular so trying to explain findings that cannot really be explained by such a dataset like why patients with metastatic disease get surgeries
Comments on the Quality of English Language
need some editing
Reviewer 3 Report
Comments and Suggestions for Authors
Congratulations on a very interesting paper that shows light and brings some surprises on pancreatoblastoma treatment outcomes. I have some questions and comments regarding the presented data:
1. Were there any hypothesis on why these patients were treated more frequently in academic medical centers? My guess is that this migh have been the case because of the rarity of the disease.
2. Can you tell on how frequent pancreatoblastoma diagnosis was preoperatory?
3. It really intrigues me why pancreatoblastomas more frequently underwent surgery compared to PDAC, and it surprised me to know that there were no T4 pancreatoblastomas, and that the smaller median tumor size compared to PDAC. One would imagine that given the aggressively nature of the disease, this would not be the case. Do you have any comments on this?
4. It called my attention the high 90-day postop mortality of pancreatoblastoma patients, compared to PDAC. Any comments on this?
5. The median survival of respected pancreatoblastoma patients really surprised me. Certainly, surgery is more pivotal for pancreatoblastoma compared to PDAC.
In conclusion, congratulations on an original paper that, at least in my case, showed unexpected results.
Round 2
Reviewer 2 Report
Comments and Suggestions for Authors
i appreciate the authors revisions